# Enhancing Breast Cancer Treatment Using a Combination of Cannabidiol and Gold Nanoparticles for Photodynamic Therapy

**DOI:** 10.3390/ijms20194771

**Published:** 2019-09-26

**Authors:** Dimakatso R. Mokoena, Blassan P. George, Heidi Abrahamse

**Affiliations:** Laser Research Centre, Faculty of Health Sciences, University of Johannesburg, P.O. Box 17011, Johannesburg 2028, South Africa; drmokoena2@gmail.com (D.R.M.); blassang@uj.ac.za (B.P.G.)

**Keywords:** photodynamic therapy (PDT), gold nanoparticle (AuNP), cannabidiol (CBD), breast cancer therapy

## Abstract

Indisputably, cancer is a global crisis that requires immediate intervention. Despite the use of conventional treatments over the past decades, it is acceptable to admit that these are expensive, invasive, associated with many side effects and, therefore, a reduced quality of life. One of the most possible solutions to this could be the use of gold nanoparticle (AuNP) conjugated photodynamic therapy (PDT) in combination with cannabidiol (CBD), a *Cannabis* derivative from the *Cannabis sativa*. Since the use of *Cannabis* has always been associated with recreation and psychoactive qualities, the positive effects of *Cannabis* or its derivatives on cancer treatment have been misunderstood and hence misinterpreted. On the other hand, AuNP-PDT is the most favoured form of treatment for cancer, due to its augmented specificity and minimal risk of side effects compared to conventional treatments. However, its use requires the consideration of several physical, biologic, pharmacologic and immunological factors, which may hinder its effectiveness if not taken into consideration. In this review, the role of gold nanoparticle mediated PDT combined with CBD treatment on breast cancer cells will be deliberated.

## 1. Introduction

Cancer has been declared the second foremost source of death worldwide. According to the World Health Organization (WHO), cancer is accountable for roughly 70% of deaths in low to middle-income countries [1]. Breast cancer, on the other hand, is the most lethal cancer among women globally and the second most common cancer after lung cancer. It is usually initiated from the mammary ducts or the lobular ducts of the breast [2,3]. Generally, breast cancer tumours are benign and not malignant, which means they can be surgically removed when detected early. When the tumour is malignant, it usually spreads via the blood circulatory system or the lymphatic system, which complicates the treatment [4]. Depending on the grade of the tumour, the treatment options would most likely include surgery followed by chemotherapy, radiation therapy or both.

Breast cancer is classified into three grades. The 1st grade is defined as a slow growing tumour that is unlikely to metastasise into other parts or organs of the body. The 2nd grade is classified as a moderately differentiated tumour compared to the normal cells, and it is between the 1st and 3rd grades. The 3rd grade is classified as a fast growing cancer whose cells are completely differentiated from the normal cells of origin. As much as the conventional therapies have become the golden standard for the past decades, their side effects and reduced survival rate cannot be ignored [4,5,6]. Therefore, less invasive yet effective forms of cancer therapies are desirable, such as cannabidiol (CBD) therapy, photodynamic therapy (PDT) or both administered concurrently.

## 2. Cannabidiol

One of the promising possible cancer therapies includes combination therapies with cannabinoids. Apart from its recreational use, *Cannabis sativa* has been used as one of the vital components in herbal remedies for many centuries. Recently it has been observed that apart from its psychoactive properties, *Cannabis* has some important derivatives called Cannabinoids, whose properties play a role in healing and cancer eradication. The Cannabinoids utilise the endocannabinoid system in cells, in which they bind to the endocannabinoid receptors, and thus initiate the metabolism of enzymes, which influence different physiological and pathological processes [7]. In cancer, the most basic reason for the use of Cannabinoids involves alleviation of pain, reduction of nausea and stimulation of appetite. However, recent research has shown that Cannabinoids have anti-tumour effects, which induce cell death pathways, cell growth arrest and tumour angiogenesis invasion and metastasis inhibition.

This discovery marked the beginning of broad investigations relating to the probable antitumour properties and symptom control of cannabinoids in cancer patients [8,9]. One of the earliest studies describing its antineoplastic properties was published in 1975 [10]. Recently, the cannabinoids associated with antitumour properties include: cannabinol (CBN), ∆9-tetrahydrocannabinol (THC), ∆8-THC, cannabidiol (CBD) and cannabicyclol (CBL). Currently, one of the trending cannabinoids includes cannabidiol (CBD) [11,12,13]. As much as CBD has been linked with anti-tumour properties, its precise mechanism is still a mystery to be uncovered; thus, more research on its activity is warranted [13,14].

Jeong and colleagues concluded that CBD induced apoptosis in colorectal cancer cells by Noxa-and-ROS-dependent pathways. Their study reveals that CBD caused an increase in mitochondrial reactive oxygen species (ROS) synthesis and endoplasmic reticulum stress, which also caused an activation of Noxa production. The activation of Noxa resulted in the subsequent stimulation of the caspases, thus resulting in apoptosis (Figure 1). They further concluded that their results have re-established the effects of CBD and its role as a reliable and innovative anticancer drug [14,15]. A study done by Yasmin-Karim and colleagues explored the potential of combination approaches employing CBD with radiotherapy (RT) to enhance the therapeutic efficiency for pancreatic and lung cancers. Their experimental results exhibited greater effective tumour cell death when using the combination of CBDs with RT in vitro [15]. Table 1 summarises some of the cannabinoids associated with antitumour properties.

## 3. Photodynamic Therapy (PDT)

Photodynamic therapy is a form of light therapy that uses visible light, a photosensitiser (PS) and molecular oxygen to destroy cancer cells and pathogenic bacteria [21]. Unlike conventional therapies, PDT is non-invasive and selectively cytotoxic to malignant cells, and it can significantly improve the quality of life and prolong the cancer survival rate. One of the major side effects presented by conventional cancer treatments is the weakened immune system, which makes PDT a more attractive form of therapy as it causes direct tumour cell death by apoptosis, necrosis and autophagy [22,23].

In PDT, the PS is distributed directly on the tumour site or systematically via the vascular system. Following the localisation of the PS, light at specific wavelength is applied in the presence of molecular oxygen. This reaction is followed by the production of ROS, which ends in oxidative damage of the intracellular elements within the cell, thus leading to cancer cell destruction [24,25] (Figure 2). When PDT targets the vascular system of the tumour, it results in haemostasis, vessel constriction and breakdown. This leads to a decreased oxygen and nutrient supply to the tumour, which eventually results in tumour cell death [26]. Furthermore, some PSs force tissue demolition and activation of acute inflammatory processes alongside a substantial anti-tumour immune response [27]. The effectiveness of PDT is highly depended on the choice of PS, which is mostly chosen on the basis of its solubility and selectivity.

### 3.1. Photosensitisers (PS)

Generally, a PS is a non-toxic dye that can never result in any damage to the cells until stimulated by light within a particular wavelength, ideally light within the red and near infrared spectral region [21]. Upon exposure to light of a specific wavelength, the PS molecule absorbs the photons of light, thus being activated from its ground state to its active excited singlet state (Figure 2) [28]. However, upon constant light exposure, the PS in the excited singlet state goes through an intersystem while moving to its excited triplet state. At this point, the PS will either combine with biomolecules in reduction reactions that yield free radicals or other radical species that react with oxygen to produce ROS (Type I). On the other hand, it can directly react with molecular oxygen in its ground state to produce singlet oxygen, which is highly toxic (Type II) [28,29].

Photosensitisers that are triggered with wavelengths near the tissue absorption range (red region) are ideal for therapeutic purposes. It is vital for PSs to retain a high triplet quantum yield for ROS synthesis upon irradiation [21]. A number of PSs have been observed for application in the PDT of most cancers, including breast cancer. Different PSs have been indicated for successful cell death induction in a variety of different cancers (Table 2). Among the PSs, Hypericin is one of the naturally occurring PSs that have been used in PDT.

Hypericin forms a component of St. John’s Wort (Hypericum perforatum), and is often used as an anti-depressive agent. In PDT, Hypericin responds to light activation between the ranges of 545 and 590 nm [30]. Hypericin triggers the phagocytosis of the cancer cell by exposing the damage associated molecular patterns, and it does that by localising in the endoplasmic reticulum (ER) of cells, which is active in the stimulation of immunogenic cell death. In other terms, it awakens the innate immune system and phagocytosis of destroyed cancer cells [31,32]. Hypericin can cause cancer cell death via the apoptotic pathway, autophagy or the necrotic pathway, which renders it an ideal PS of choice. It is rapidly removed by normal healthy cells and, hence, can be used as a fluorescent marker for diagnostic imaging [33,34,35]. However, because of its hydrophobicity, it is almost impossible to apply therapeutic doses of Hypericin [21,36]. The PS tends to aggregate in aqueous milieu, resulting in low bioavailability and the loss of photodynamic activity [37]. Nevertheless, not all good PSs are water soluble, and hence, nano-conjugating them to organic and inorganic nanoparticles is a possible approach to fulfil the necessities of an ultimate PDT system [23,24,25,26,38].

### 3.2. Advances in the Field of PDT

Numerous therapeutic drugs struggle with different limitations including low aqueous solubility and a shortened half-life. This necessitates the need to design a drug delivery system that will improve the half-life and the drug target specificity. Over the years, conventional drug delivery systems have displayed their efficacy to some extent, as these systems are currently overwhelmed with the challenge of effective drug delivery for many other drugs today. Nanoparticle aided drug delivery offers a podium to adjust the simple properties of drug molecules by improving the half-life, solubility, biocompatibility and drug release characteristics. While a number of nanoparticle centred products have been made commercially available, more and more are being developed, and some are already going through clinical and preclinical trials [62].

One of the currently used nanoparticles is liposomes. Their use has been expanded subsequently after their discovery in 1964. Liposomes can be made up of from a number of different lipids, including those that are synthetic and those that are naturally occurring. The most commonly used lipid is phosphatidylcholine. Phosphatidylcholine is a low cost type of lipid that is also a main constituent of cell membranes. Liposomes are categorised on the basis of their synthesis method, conformation, size and shape. In drug delivery, they enhance the pharmacokinetics and pharmacodynamics, reduce cytotoxicity and improve selectivity and drug delivery [63,64].

## 4. Hypericin and Gold Nanoparticle

In the beginning of nanotechnology in 1959, Laureate Richard Feynman proposed, “There is plenty of room at the bottom,” and it was only a decade later that Professor Norio Taniguchi devised the term Nanotechnology [65]. Principally, Nanotechnology is the study of the physics, chemistry, biology and technology of nanometre-scale objects or nanoparticles. Nanoparticles are all particles signified by the nanoscale dimension, measuring 100 nanometres or less. Because of their unique size-dependent physical and chemical properties, nanoparticles have been extensively studied and have valuable applications in many areas of study including biology and medicine [65,66].

In PDT, these nanotechnological innovations have very significant applications. Nanoparticles have many physical, chemical and biological properties that enable them to perform unique interactions with biological systems at the molecular level and, hence, their vital role in the cancer therapeutics. Due to their biocompatibility, solubility, improved half-life and easy release characteristics, a number of nanoparticle based therapeutic products have been used recently. Table 3 summarises the nanoparticles currently used and their properties [62]. Among the different types used in biology, AuNPs are the most commonly used because of their wide availability and unique properties [29,65].

AuNPs have the highest potential to cross the endothelial membranes of cancer cells during therapy. In PDT, the most important factor to overcome is how to get the photosensitiser internalised in cells so that upon excitation, there is enough production of ROS in the cells. Many PSs, including Hypericin, are hydrophobic and poorly water-soluble, which limit their ability to move across membranes. Thus, a carrier system is essential to improve the effectiveness of PDT using Hypericin. Tagging AuNPs to Hypericin, therefore, enhances their movement across membranes by increasing their aqueous solubility, bioavailability and stability [65]. This is a promising strategy for the targeted delivery of Hypericin to breast cancer cells with increased efficiency of PDT for eradicating tumour cells [66].

In vivo, AuNPs also have the ability to escape recognition and possible interference by the immune system. Physiologically, the immune system recognises all foreign substances, including therapeutic drugs, as invaders. When injected intravenously or intramuscularly, PSs can potentially be affected by the immune reaction of the host. This results in either denaturation of the drug or any other interference of the pharmacodynamics of the drug in the body. Nanoparticles, including AuNPs, mimic biological components in the body and, therefore, remain undetected by the immune system barriers, creating an excellent delivery system for therapeutic drugs [46]. AuNPs are large enough to be retained within the systemic circulation and are able to escape opsonisation by the reticulo-endothelial system. Studies by Stuchinskaya and colleagues [46] demonstrated an improved PDT drug delivery system for lung cancer using AuNPs combined with PSs. In this study, AuNPs appeared to be responsible for the preservation and maintenance of the PS. The nanoparticles aided in obscuring the PS from biological barriers and enzymes in vitro and, hence, increased the cellular uptake drug load, yielding better ROS production [46].

Additionally, the ability to escape immune recognition, together with the enhanced permeability and retention effect, enables nanoparticles to stay longer in the systemic circulation, further increasing the therapeutic potential [81]. Portilho and colleagues noted and reported that conjugates of PSs and AuNPs as cancer drug delivery applications have a better PDT efficiency, with higher triplet lifetimes than unbound PSs [82].

In PDT, each PS molecule is responsible for the production of ROS upon irradiation. It is therefore very important to ensure increased concentration of PSs in the cells in order to increase the cytotoxic effects. The small dimensions (i.e., size, structure and core) of AuNPs allow them to accumulate in cells, especially the poor lymphatic drainage tumour cells that are the target [83,84,85]. AuNPs can be surface functionalised with diverse groups [66,86]. Their large surface area enables them to accommodate many different functional groups and to form different forms of the same particle. With this surface area, many Hypericin molecules can be coupled to AuNPs to increase the concentration of the PS in the cells. Apart from being conjugated to PSs, AuNPs can also be coated with mono-specific antibodies, and also deoxyribonucleic acid (DNA), ribonucleic acid (RNA) or protein, further enhancing drug delivery and specificity [87,88,89]. In spite of the understanding of nanoparticle interaction with biological components, there are still many gaps in the full comprehension of biological processes involved in the interaction. Future studies are required to improve the overall understanding of the physicochemical parameters of nanoparticles in vivo [90].

## 5. Combination Therapy of Cannabidiol and Photodynamic Therapy

Because of the complexity of cancer cells, most therapies administered alone are not capable of causing maximum cancer eradication. This is due to cancer cells’ ability to adapt to changes in the microenvironment rendering mono-therapeutic approaches ineffective. Therefore, to maximally eradicate cancer cells, combining the anticancer properties of CBD with the cytotoxic properties of PDT is a commendable approach. Complete tumour eradication exclusively using CBD alone remains challenging [91]. CBD targets the G-protein coupled cannabinoid receptors (CB1-R and CB2-R), which are upregulated in cancer. In breast cancer for example, CB1-R is minimally expressed while CB2-R expression is high, and the binding of CBD to the CB2-R prevents tumour progression and metastasis [92]. PDT, in contrast, causes cancer cell death via production of cytotoxic oxygen species, which, when combined with the initial effect cause by CBD, makes complete cancer cell eradication conceivable. Some studies have attempted to improve the efficacy of PDT by combining it with ultrasound and sonosensitive agents [93], and others have used it with radioactive materials. A study done by Yang and colleagues (2017) proved that in order to improve the suppression of tumour cells, a more synergistic combination therapy is ideal, as it targets more than one cell killing pathway, thus ensuring complete tumour eradication [94]. In their study, they made use of two PSs, which are known to target cancer cell destruction in two different pathways: one via apoptosis and the other via necrotic cell death. Among the current combination studies, none has attempted combining PDT with a more natural therapy, such as CBD. Hypothetically, this type of combination could achieve complete cancer eradication in breast cancer, as hypothesised in Figure 3 below.

Both PDT and CBD seem to induce cancer cell death by apoptosis, necrosis and autophagy [95]. However, the exact mechanism in which each cell death pathway is activated is not clear. Apoptosis is described as a ‘programmed cell death’ activated via the mitochondrial stress (intrinsic) or via the death receptor activation (extrinsic pathway). Necrosis is described as cell death resulting from physical or chemical distress [96,97]. In autophagy, a cell experiences a self-destructing course resulting in the degradation or recycling of the intracellular elements [98]. Although it is tricky to understand the modes of cell death, it is still fundamental to interpret the differences for the improvement and delivery of optimal cancer treatments regimes [98,99,100].

The advantage of using PDT for cancer treatment is that the PS is selectively absorbed by tumour cells when administered into the body. There are many reasons for the selective absorption of the PS by tumour cells, including the dense vacuolisation of tumour tissue, which increases the surface area for PS uptake [100], the increased permeability of cancer cell membranes and the presence of increased Low Density Lipoproteins (LDL) receptors, which increases PS uptake via the endocytosis of LDL-PS complexes [101]. This results in the PS having a higher affinity for cancer cells than normal cells. When the PS is excited after irradiation with the corresponding wavelength of light, chemical interactions that follow the two pathways occurs, as explained in Figure 2. There is no clear explanation as to what factors transpire for the occurrence of which reaction, but it is believed that these reactions occur independent of each other and can occur concurrently.

With either of the reactions happening, the ultimate cytotoxicity of the cells occurs by either, or a combination of, necrosis, apoptosis and autophagy. The final pathway taken depends on where the PS localises in the cell. When a PS accumulates in the cytoplasm or in cytoplasmic organelles, the activation of apoptosis occurs, and the cells are killed via any of the different apoptotic pathways (intrinsic and extrinsic). Accumulation of the PS on the cellular membrane injures the cell membrane integrity causing a necrotic cell death. [95]. In the body, PDT also activates the host antitumour immune response and can cause destruction of the tumour vasculature when the PS binds to the dense network of vessels supplying the nutrients to the tumour [102]. When this occurs, cells are killed by the issuing ischemia and via necrosis. With little scientific explanation to the molecular mechanisms of autophagy, it is not clear as to how PDT causes autophagic cell death. However, PDT has also demonstrated the death of cells via autophagy in cells that are apoptosis deficient. This occurs when PDT damages many of the proteins that prevent autophagy and also the organelles that are directly involved in the process [95].

## 6. Conclusions

Combining PDT with CBD will allow a substantial advance in the treatment of breast cancer and other types of cancers. Without a doubt, this combination will result in reduced side effects and toxicity to normal cells, and it will improve the patient’s quality of life. Bearing in mind the above, PDT on its own is already receiving an augmented prominence as a form of therapy, not only between clinicians, but also among patients. CBD is also a promising anticancer agent, and has been linked to a number positive effects for cancer patients, so its impact cannot be ignored. Taken together, PDT and CBD are promising means for hindering breast cancer progression and development. However, to improve the efficacy of these therapies, more investigation of the molecular pathways linked with both therapies is required.

## Figures and Tables

**Figure 1 ijms-20-04771-f001:**
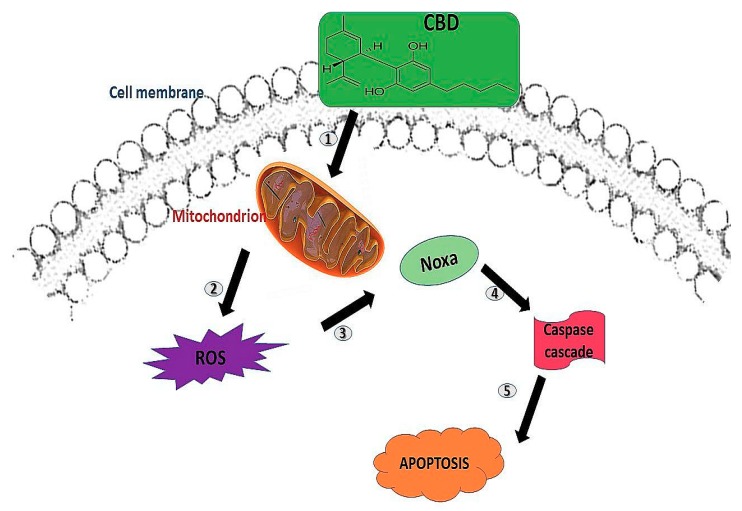
The proposed mechanism of cannabidiol (CBD) in inducing apoptosis. (1) CBD binds to the central and peripheral receptors on the cell membrane; (2) this results in mitochondrial stress and reactive oxygen species (ROS) production (3) leading to Noxa activation, and thus (4) the caspase cascade activation, which results in (5) apoptosis.

**Figure 2 ijms-20-04771-f002:**
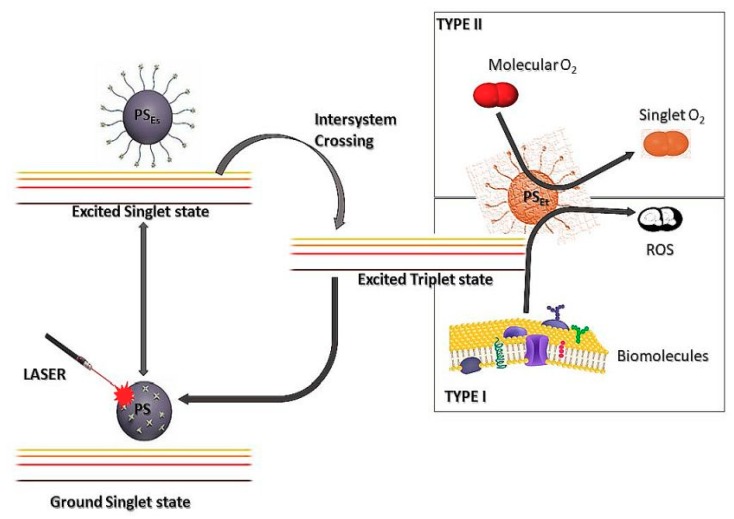
A schematic Jablonski diagram indicating the type I and type II reactions that cause cytotoxicity after activation of the photosensitiser (PS) with light of a specific wavelength.

**Figure 3 ijms-20-04771-f003:**
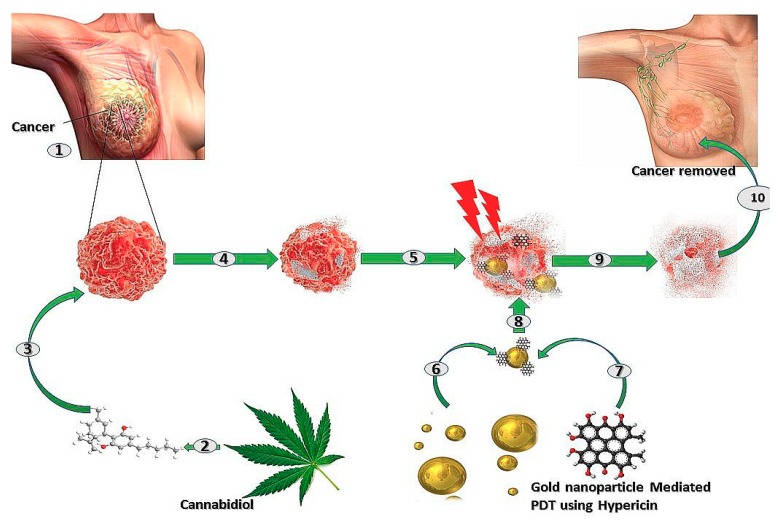
Combination therapy using cannabidiol and PDT to effectively treat breast cancer. (1) A lump of tumour in the right breast of a female. (2) *Cannabis* leaf can be used to extract Cannabidiol, (3) which, when injected in the body, binds to the Cannabidiol receptors on the tumour cells, (4) inducing cell death via activation of apoptosis. (5) Cells of the tumour start dying but Cannabidiol treatment alone is not efficient. (6) AuNPs and (7) Hypericin can be conjugated to form (8) a AuNP-Hypericin PS, which, when used in the PDT of the Cannabidiol treated tumour cells by exposure to light at a wavelength of 594 nm, (9) induces cytotoxicity of the cancer cells further enhancing cell death. (10) Ultimately, with combined therapy, there is maximum eradication of the tumour, restoring normal breast tissue.

**Table 1 ijms-20-04771-t001:** Antitumour properties of some of the *Cannabis* derivatives.

Cannabinoid	Therapeutic Properties	Cell Death Mechanism	Reference
Cannabidiol (CBD)	Reduced cancer cell viability, inhibits cell proliferation and invasion.	Apoptosis and autophagy.	[16,17,18]
Tetrahydrocannabinol (THC)	Decreased cancer cell proliferation.	Induction of p8-ATF-4TRIB3 pro-apoptotic pathway	[16,17,18,19]
CBD and THC combined	Synergistic inhibition of cellular proliferation.	Cell cycle modulation, ROS synthesis, apoptosis and caspase activities.	[20]
Cannabigerol (CBG)	Decreased cell viability	Induced apoptosis (Intrinsic apoptotic pathways)	[16]

**Table 2 ijms-20-04771-t002:** Different types of photosensitisers and their application in different cancers.

Category	Photosensitiser	Wavelength Used	Type of Cancer	References
Tetrapyrrole structures	Porphyrins	630 nm	Hepatocellular cancer, leukaemia and nasopharyngeal carcinoma	[21,39,40,41]
	Chlorins	650–700 nm	Colon, prostate, bronchial and oesophageal cancers	[42,43,44]
	Phthalocyanines	640–690 nm	Cutaneous and subcutaneous lesions for many solid tumours including breast, cervical, melanoma, oesophageal and colon	[45,46,47,48]
	Bacteriochlorins	700–800 nm	Colon	[49]
Synthetic dyes	Rose Bengal	530–540 nm	Breast and oral	[21,50]
	Phenothiazinium salts	630–670 nm	Fibrosarcoma	[21,51]
	Transition metal compounds		Breast and gastrointestinal carcinomas	[52]
	BODIPY dye	530–540 nm	9 cancer cell lines in vitro and an ovarian cancer cell line in vivo using a murine peritoneal cancer model	[21,53]
Natural products	Hypericin	470–570 nm	Breast cancer, ovarian cancer and colon cancer	[54,55,56]
	Riboflavin	310–700 nm	Liver cancer, colorectal and cervical	[57,58]
	Curcumin	350–450 nm	Nasopharyngeal carcinoma and breast	[59,60]
	Hypocrellins	532 nm	Cervical and gastric	[61]

**Table 3 ijms-20-04771-t003:** Nanoparticles and their applications.

Nanoparticle	Description	Application	Reference
Polymeric nanoparticle	Biodegradable, biocompatible and non-toxic colloidal particles ranging between 1 and 1000 nm in size. Used to carry pharmaceutical drugs by adsorption, conjugation with a linker or encapsulation.	Drug delivery tissue engineering and gene delivery.	[67,68]
Polymeric Micelles	Amphiphilic co-polymers that gather forming a micelle with a hydrophobic core and a hydrophilic shell.	Multidrug co-delivery and cancer treatment.	[69,70]
Dendrimers	Three dimensional, outward emerging, well defined structures with systematic patterns and recurring units. Highly functionalised polymers resembling biomolecules.	Drug and gene delivery. Limitation involves cytotoxicity to cells.	[62,71]
Liposomes	Self-assembling vesicular colloidal structures with a membrane composed of lipid bilayers.	Controlled drug loading and release.	[63,72,73]
Viral based nanoparticles	Multivalent, self-assembled protein cages. Synthesised mostly from naturally occurring plant viruses.	Gene therapy and drug delivery.	[74,75]
Carbon nanotubes	Graphene sheets rolled up into cylinders.	Drug and protein delivery and gene therapy.	[76,77,78,79]
Quantum dots	Semiconducting Nanocrystals with a size range of 2–10 nm	Drug delivery in photo-thermal and photodynamic therapy combination properties, imaging and immunosensing.	[79,80]

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
