# Peer review of "Enhancing Breast Cancer Treatment Using a Combination of Cannabidiol and Gold Nanoparticles for Photodynamic Therapy"

_ijms, 2019, doi:10.3390/ijms20194771_

Round 1

Reviewer 1 Report

Authors review the development of photodynamic therapy (PDT) on breast cancer including some photosensitizers, drug-carriers, nanoparticles and the combination therapy. In this review, “combination therapy of cannabidiol and photodynamic therapy” is only one small section of the manuscript. The title of this manuscript “Enhancing breast cancer treatment using a combination of conjugated Cannabidiol and Gold nanoparticles for Photodynamic Therapy” does not truly reflect the content. In addition, the mechanism of PDT includes type I and type II. The label of type I and type II should be exchanged in Figure 2.

Reviewer 2 Report

The manuscript by Dimakatso et al. reviews current literature in the photodynamic approach to tumor treatment, undoubtedly a hot topic in drug delivery. Pharmacologic effect of cannabinoids, photosensitizers and nanoparticles are covered. Although the review has the potential to contribute to the field, I think that some aspects should be better described and covered before the review is publishable. In details:

1) the interaction between immune system and nanoparticles is somewhat overlooked. While it is true that some nanoparticles can improved biodistribution, the authors should better described the fact that, especially after repeated administrations, nanostructured systems can be easily recognized and cleared. This is a relevant point that currently represents one of the major hurdles to the wide employ of nanoparticles in clinics.

2) it is not clear from the manuscript the correlation (besides the obvious comined effect) between a pharmacologic actor (cannabinoid) and a photodynamic agent. Since the review focuses on the combined use of these two components, a more detailed description of the possibility to administer them combined would be expected, along with the potential issues of appropriately dosing the right concentration of the drugs (i.e. cannabionids vs. nanoparticles).

3) there are numerous other nanoparticles that appeared recently in the literature trying to overcome the two above mentioned limitations, with particular respect to the possibility to perform co-delivery of chemiotherapic and photosensitizing effect, and to the capability to avoid opsonization and accelerated clearance. The authors might want to give an overview of such compounds.

If the authors can address these issues, I think that the review could represent a good contribution to current literature.

Round 2

Reviewer 2 Report

the authors addresses my concerns. The manuscript is now suitable for publication